

# Impact of demographics on human gut microbial diversity in a US Midwest population

Jun Chen[1,2], Euijung Ryu[1], Matthew Hathcock[1], Karla Ballman[1,3], Nicholas Chia[1,2,4,5], Janet E Olson[1] and Heidi Nelson[2,4]

[1] Department of Health Sciences Research, Mayo Clinic, Rochester, Minnesota, United States
[2] Microbiome Program, Center for Individualized Medicine, Mayo Clinic, Rochester, Minnesota, United States
[3] Division of Biostatistics and Epidemiology, Weill Medical College of Cornell University, New York, New York, United States
[4] Department of Surgery, Mayo Clinic, Rochester, Minnesota, United States
[5] Department of Physiology and Biomedical Engineering, Mayo Clinic, Rochester, Minnesota, United States

Corresponding author
Heidi Nelson, nelsonh@mayo.edu

## ABSTRACT

The clinical utility of microbiome biomarkers depends on the reliable and reproducible nature of comparative results. Underappreciation of the variation associated with common demographic, health, and behavioral factors may confound associations of interest and generate false positives. Here, we present the Midwestern Reference Panel (MWRP), a resource for comparative gut microbiome studies conducted in the Midwestern United States. We analyzed the relationships between demographic and health behavior-related factors and the microbiota in this cohort, and estimated their effect sizes. Most variables investigated were associated with the gut microbiota. Specifically, body mass index (BMI), race, sex, and alcohol use were significantly associated with microbial $\beta$-diversity ($P < 0.05$, unweighted UniFrac). BMI, race and alcohol use were also significantly associated with microbial $\alpha$-diversity ($P < 0.05$, species richness). Tobacco use showed a trend toward association with the microbiota ($P < 0.1$, unweighted UniFrac). The effect sizes of the associations, as quantified by adjusted $R^2$ values based on unweighted UniFrac distances, were small (< 1% for all variables), indicating that these factors explain only a small percentage of overall microbiota variability. Nevertheless, the significant associations between these variables and the gut microbiota suggest that they could still be potential confounders in comparative studies and that controlling for these variables in study design, which is the main objective of the MWRP, is important for increasing reproducibility in comparative microbiome studies.

Subjects Microbiology, Gastroenterology and hepatology, Public health, Statistics
Keywords Microbiome, Target population, Effect size, Demographics, Microbial diversity

## INTRODUCTION

Humans are populated by a vast number of microbes. It is estimated that bacterial cells associated with the human body outnumber human cells by a factor of 10 (*Cho & Blaser, 2012*). These microbes and their genetic content constitute the human

microbiome (*Cho & Blaser, 2012*). The gut microbiome alone accounts for more than three million genes, more than 100 times the number of human genes (*Qin et al., 2010*). The microbiome has both digestive and metabolic functions (*Cani & Delzenne, 2009*) and plays an important role in the development of the host immune system (*Round & Mazmanian, 2009*). Extensive evidence shows that a core microbiome is responsible for maintaining a healthy state; any significant deviation may affect an individual's risk of disease (*Sartor, 2004*; *Ley et al., 2006*; *Huse et al., 2012*; *Human Microbiome Project Consortium, 2012*; *Zhang et al., 2015*).

Recognition of the microbiome's importance to human health and disease has inspired a surge of studies evaluating changes in the gut microbiota in various conditions, including cancer, inflammatory diseases, and obesity (*Turnbaugh et al., 2006*; *Qin et al., 2012*; *Scher et al., 2013*; *David, 2013*; *Schubert et al., 2014*; *Abreu & Peek, 2014*). To adequately evaluate the changes associated with these conditions, a truly representative control group must be used. However, the definition of an ideal control group is complicated by the complexity of the microbiota and its extensive inter- and intra-individual variation. Historically, it has been presumed that young adults who have no identified and/or treated medical conditions are ideally suited for establishing normative laboratory values. This approach reduces confounding influences and provides an opportunity to consider what the structure of a "healthy" adult microbial community might look like. Therefore, attempts have been made to generate and characterize cohorts of "healthy" individuals, with the intention of developing standard reference microbiomes for use in future studies. The Human Microbiome Project (HMP) and the MetaHIT represent two such large-scale endeavors (*Qin et al., 2010*; *Gevers et al., 2012*; *Foxman & Rosenthal, 2013*). However, "healthy" individuals may not accurately represent an epidemiologic population with diverse health conditions. Moreover, the distribution of demographic variables in such a group may be different from that of the epidemiologic population, creating the potential for confounding in comparative studies.

In epidemiologic studies of US Midwestern populations at our institute, the Mayo Clinic, normative values are generated, validated, and used to help guide diagnostic and therapeutic purposes. These normative values typically apply to broad populations of people with diverse health conditions, such that a lab value outside the normative range suggests a specific condition or a set of conditions. Here, we apply this approach to the microbiome, offering a broad Midwestern Reference Panel (MWRP) for use in disease association studies of Midwestern populations.

The objectives of the current study were twofold. First, we aimed to create a representative epidemiologic sample from the Midwestern United States, excluding subjects with conditions that would directly impact the gut microbiota, such as gastrointestinal (GI) disorders. Second, we aimed to study the effect sizes of demographic and health behavior–related variables, including sex, age, race, BMI, alcohol use, and smoking habits, on the gut microbiota. Thus, in addition to introducing a truly representative cohort for the Midwestern United States population, this study reveals which demographic and health behavior–related variables have the greatest effect on the gut microbiome and, therefore, should be given special attention in study design and the interpretation of results.

## MATERIALS AND METHODS

### Biobank/consent

Subjects were selected from the Mayo Clinic Biobank, an institutional resource composed of biological specimens accompanied by clinical data obtained from patient medical records and patient-provided risk factor data (*Olson et al., 2013*). Written informed consent was provided by all individuals enrolled. The study was approved by the Institutional Review Board of the Mayo Clinic (#13-003694).

### Subject selection/questionnaire

As described below, representative fecal samples were chosen according to sex, age, race, body mass index (BMI), alcohol use, and tobacco use. A survey accompanied the invitation to participate in the study and inquired about bowel symptoms, such as diarrhea, constipation, blood in the stool, and nausea or vomiting; current medication use; current supplement use; antibiotic use in the past 2 weeks; current smoking status; current alcohol intake; current weight; and cancer history.

To define a sample truly representative of the Midwestern United States (Minnesota, Iowa, North Dakota, South Dakota, and Wisconsin), we estimated the distributions of selected behavioral risk factors among subjects included in the Center for Disease Control's 2011 Behavioral Risk Factor Surveillance Survey (BRFSS; http://www.cdc.gov/brfss/annual_data/annual_2011.htm). Mayo Clinic Biobank participants were then selected based upon BRFSS distributions of age, sex, race, BMI, smoking status, and alcohol intake. Two rounds of subject selection were employed for this study. The first round took place in the summer of 2013. During this round, 380 subjects (ages 20–49 years) were selected. In the summer of 2014, a second group of 267 patients (ages 50–79 years) was selected. Stool samples were collected from 25% of the samples who consented in the first round and 31% of the subjects in the second round.

The final, combined sample was chosen on the basis of age (20–29, 30–39, 40–49, 50–59, 60–69, and 70–79 year groups), sex (10 males and 10 females in each age group), race (white or non-white), BMI ($<30$ kg/m$^2$ or $\geq 30$ kg/m$^2$), smoking status (current or former/never), and alcohol use (yes or no). The proportions of all inclusion criteria were chosen to reflect the underlying population in the Upper Midwestern states, as estimated by data from the BRFSS. Exclusion criteria, assessed based on survey information completed at the time of enrollment into the study, included use of medications, non-GI cancers, use of any antibiotic within the past 2 weeks, or any known GI disease or symptoms including, but not limited to, inflammatory bowel disease, irritable bowel syndrome, esophageal cancer, intestinal cancer, constipation, diarrhea, *Clostridium difficile* infection, and celiac disease/sprue.

### Collection method

The collection process was reviewed in detail with study participants, and a stool collection kit was provided to each subject. Stool samples were collected by the subjects and returned to Mayo Clinic Rochester within 24 hours of passing. Specimens were stored at −80 °C until DNA extraction.

## Sample preparation and sequencing

Fecal DNA was extracted using the PowerSoil kit (MoBio, Carlsbad, CA, USA) according to the manufacturer's instructions. Genomic DNA was used as a template for the polymerase chain reaction (PCR), with 0.3 μM V3–V5 barcoded primers (*Caporaso et al., 2012*) targeting 357F and 926R of the bacterial 16S gene (5′AATGATAC GGCGACCAC CGAGATCTACACTATGGTAATTGTCCTACGGGAGGCAGCAG3′ and 5′CAAGCAGAAGACGGCATACGAGATNNNNNNNNNNNNAGTCAGTCAGCCCCGT CAATTCMTTTR AGT3′, respectively). PCR conditions were as follows: 95 °C/3 min; 35 cycles of 98 °C/30 s, 70 °C/15 s, and 72 °C/15 s; and finally 72 °C/5 min in a Bio-Rad T100 Thermocycler (Bio-Rad, Hercules, CA, USA) using Kapa Hotstart Hi-Fi DNA polymerase (Kapa Biosystems, Boston, MA, USA). PCR product sizes were verified using the Agilent TapeStation with reaction cleanup, and DNA was purified using an epMotion automated system (Eppendorf, Hauppauge, NY, USA) with the Agencourt AMPure PCR Purification System (Beckman Coulter, Brea, CA). Final quantitation was performed using a QuBit HS dsDNA kit and the QuBit 2.0 fluorimeter (Life Technologies, Carlsbad, CA, USA). Samples were pooled to equal concentration and sequenced on one lane of a MiSeq at the Mayo Genomics Facility using the MiSeq Reagent Kit v2 (2 × 250 reads, 500 cycles; Illumina Inc., San Diego, CA, USA). Pre-processed sequence files were then processed via the IM-TORNADO bioinformatics pipeline with the default parameter settings to form operational taxonomic units (OTUs) (*Jeraldo et al., 2014*). IM-TORNADO uses paired-end reads to form OTUs, and was shown to be more sensitive than methods using single-end reads based on synthetic mock community studies.

## Statistical analyses

Statistical analysis proceeded in two steps. First, overall associations between demographic and health behavior–related variables and the microbiota were investigated. Next, specific associations at the level of taxa were investigated.

To perform overall association tests, we summarized microbiota data using both α-diversity and β-diversity. α-diversity reflects species richness and evenness within bacterial populations. Two α-diversity metrics, the observed OTU number and the Shannon index, were investigated. The observed OTU number reflects species richness, whereas the Shannon index measures both species richness and evenness. Before calculating the α-diversity metrics, we first replaced the observed singleton OTU count with a more robust estimate to reduce the influence of sequencing errors (*Chiu & Chao, 2015*). We then computed the α-diversity estimates at the minimum sample coverage among all samples (0.997) to standardize by sample completeness ("estimateD" in the R package iNEXT) (*Chao et al., 2014*). β-diversity reflects the shared diversity between bacterial populations in terms of ecological distance; different distance metrics provide distinctive views of community structure. Two β-diversity measures, unweighted and weighted UniFrac distances, were calculated using the OTU table and a phylogenetic tree ("GUniFrac" function in the R package GUniFrac) (*Lozupone & Knight, 2005*; *Chen et al., 2012*). The unweighted UniFrac reflects differences in community membership (i.e., the presence or absence of an OTU),

whereas the weighted UniFrac captures this information and also differences in abundance. Rarefaction was performed on the OTU table before calculating UniFrac distances.

To test the overall association based on these diversity measures, we used a regression model:

Microbiota ∼ Batch + X + X:Batch,

where microbiota is the outcome variable, summarized by the diversity measures described above, and X is the variable of interest. In the model, age and BMI were treated as continuous variables while the other variables as categorical. We included a batch variable to account for potential batch effects, since sequencing was performed in two batches, and a covariate-Batch (X:Batch) interaction term. The interaction term allowed different association strengths for the two batches. We assessed the overall effect (main + interaction effect), which is equivalent to testing the null hypothesis of no association in either batch. To assess the association with $\alpha$-diversity measures, we performed regular linear regression analysis with the likelihood ratio test of regression coefficients, as the outcome was approximately normal. To assess the association with $\beta$-diversity measures, we used the PERMANOVA procedure ("adonis" function in the R package vegan), which is a multivariate analysis of variance based on distance matrices and permutation (*McArdle & Anderson, 2001*). Significance was assessed by 1,000 permutations. Ordination plots were generated using principal coordinate analysis (PCoA) on unweighted UniFrac-based distances as implemented in R ("cmdscale" function in the standard R package).

A distance-based coefficient of determination, $R^2$ (i.e., the percentage of overall microbiota variability explained by a variable), was used to quantify the effect size of the overall association (*McArdle & Anderson, 2001*). Distance-based $R^2$ is defined as

$$R^2 = \frac{\text{tr}(HGH)}{\text{tr}(G)},$$

where tr(.) is the trace of a matrix, H is the projection matrix into the column space spanned by the corresponding variable, and G is the Gower's centered matrix, which is defined as

$$G = \left(I - \frac{11^T}{n}\right)A\left(I - \frac{11^T}{n}\right),$$

where I is an identity matrix, 1 is a vector of 1s, and $A = (a_{ij})_{n \times n} = (-d_{ij}^2/2)_{n \times n}$ is a matrix constructed using the pairwise distances $d_{ij}$. In the formula, the total variability of the microbiota was summarized using a distance metric. To quantify the overall effect size well, a good distance metric should capture the association signals. In this study, we used

the unweighted UniFrac distance, as most of the associations investigated were significant only for this metric. To account for potential overestimation due to a small sample size, we calculated an adjusted $R^2$, which is defined as

$$R_{adj}^2 = 1 - \frac{(1 - R^2)(n - 1)}{n - p - 1},$$

where n is the sample size and p is the degree of freedom of the covariate.

Next, we assessed associations between demographic and health behavior–related factors and the relative abundance of taxa. To do so, we square-root transformed relative abundance data and used a linear model adjusting for batches where appropriate. To address the non-normality of the taxa data, significance was assessed with 1,000 permutations; the F-statistic was the test statistic. False discovery rate (FDR) control was performed based on the Benjamini-Hochberg procedure to correct for multiple testing ("p.adjust" in R). To reduce the number of tests, we confined the analysis to taxa with prevalences greater than 10% and median nonzero proportions greater than 0.05%. An FDR-adjusted P-value (or Q-value) of less than 10% was considered significant. All statistical analyses were performed in R 3.0.2 (R Development Core Team, Vienna, Austria).

## RESULTS

### Gut microbiota profile of the Midwestern United States

As revealed by a comparison with BRFSS data, the distribution of important demographic variables within the MWRP cohort, which consists of 118 subjects drawn from the Mayo Clinic Biobank, generally reflects the demographic characteristics of the Midwestern United States, although alcohol users are slightly over-represented in subjects older than 50 years of age (Table 1).

Stool samples from this cohort were collected and deep sequenced. 16S rDNA–targeted sequencing yielded 110,997 reads/sample on average (range: 33,579–385,720). Clustering of these 16S sequence tags produced 1,745 non-singleton OTUs at a 97% similarity level. The distribution of OTU abundance and prevalence was typical, dominated by rare and low-abundance OTUs (Figs. 1A and 1B). An average of 402 OTUs was detected in each subject (range: 186–810). The median OTU abundance was 163 read counts per OTU (range: 2–1,456,439), and the median OTU prevalence was 11% (range: 0.8%–100%). Only 61 OTUs (3.5% of total OTUs) occurred in more than 95% of samples, indicating a relatively small core set of OTUs. The detected OTUs were classified into 11 phyla, 80 families, and 171 genera using the Ribosomal Database Project (RDP) classifier. At the phylum level, Firmicutes accounted for 48.3% of total reads and Bacteroides for 48.1% of total reads. At the family level, the most abundant families were *Bacteroidaceae* (33.76%), *Ruminococcaceae* (22.94%), and *Lachnospiraceae* (15.13%). At the genus level, the dominant genera were *Bacteroides* (33.76%), *Ruminococcus* (9.83%), and *Faecalibacterium* (9.13%). Consistent with previous studies, we observed large intersubject variability in taxa abundance (Figs. 1C and 1D) (*Human Microbiome Project Consortium, 2012*; *Flores et al., 2014*). Table 2 lists the "core" taxa (prevalence greater than 95%) identified in this study; we identified 4 core phyla, 18 core families, and 22 core

**Table 1 Comparison between subjects in the MWRP (N = 118) and the BRFSS cohort on the major demographic and health behavior-related factors.**

|  |  | MWRP | | BRFSS | |
|  |  | Number | % | % | P-value |
| --- | --- | --- | --- | --- | --- |
| Age <50 years (n = 58) | Sex, F | 30 | 51.7 | 49.3 | 0.81 |
|  | Race, W | 49 | 84.5 | 73.5 | 0.08 |
|  | BMI >30 kg/m$^2$ | 14 | 24.1 | 25.6 | 0.92 |
|  | Alcohol use (Y) | 41 | 70.7 | 60.2 | 0.13 |
|  | Smoking (Y) | 9 | 15.5 | 23.3 | 0.21 |
| Age ≥50 years (n = 60) | Sex, F | 30 | 50.0 | 53.9 | 0.63 |
|  | Race, W | 54 | 90.0 | 83.2 | 0.22 |
|  | BMI >30 kg/m$^2$ | 20 | 33.3 | 29.5 | 0.61 |
|  | Alcohol use (Y) | 39 | 65.0 | 48.4 | 0.02 |
|  | Smoking (Y) | 8 | 13.3 | 15.7 | 0.74 |

**Abbreviations:**
BMI, body mass index; BRFSS, Behavioral Risk Factor Surveillance Survey; F, female; MWRP, Midwestern Reference Panel; W, white; Y, yes.

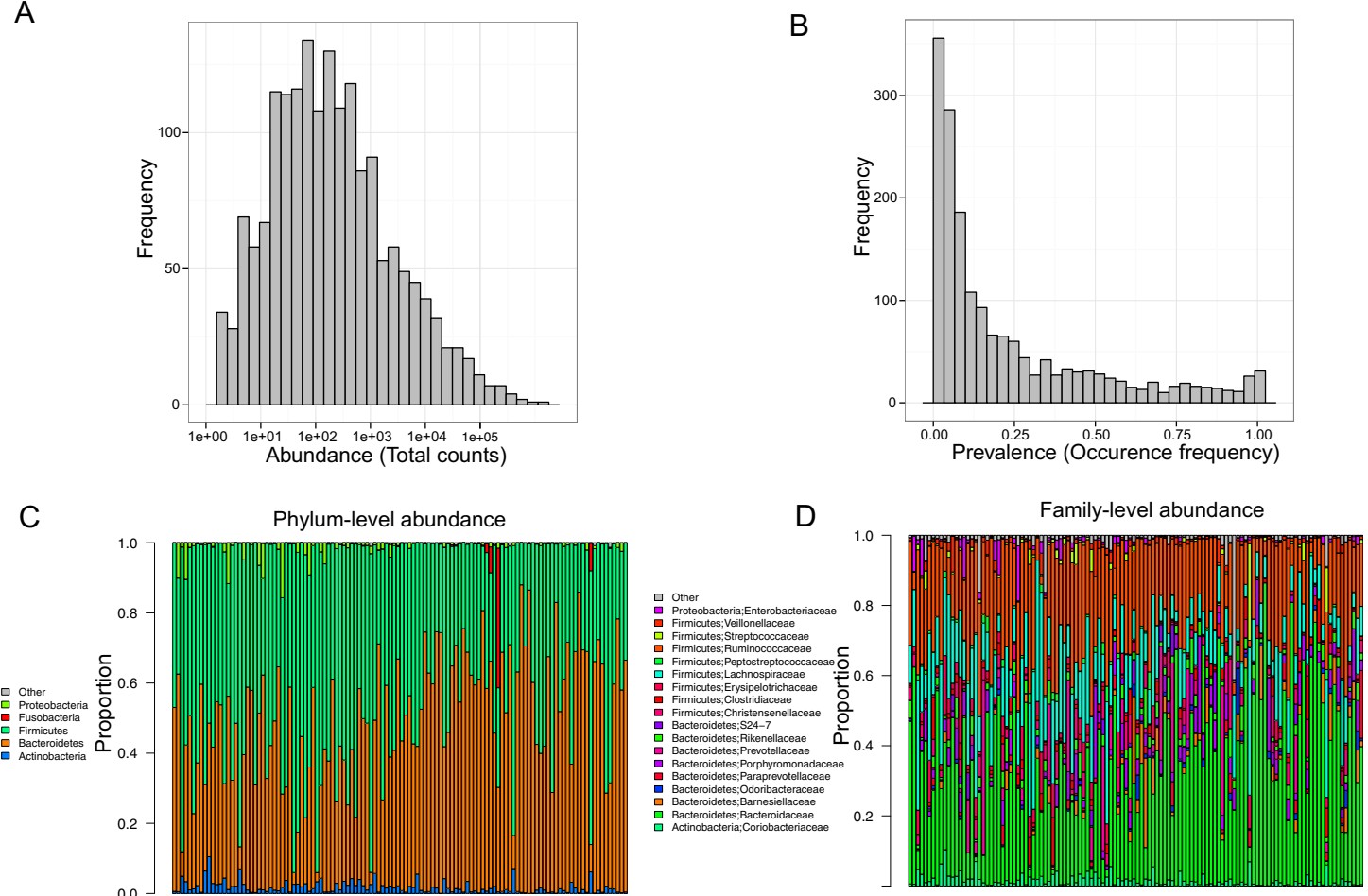

**Figure 1 Gut microbiota profile of the MWRP.** (A) Distribution of OTU abundance. (B) Distribution of OTU prevalence. (C) Relative abundance of major bacteria at the level of the phylum. (D) Relative abundance of major bacteria at the level of the family.

**Table 2 Most prevalent taxa (≥95%) at the phylum, family, and genus level identified in the MWRP.**

| Phylum | | | |
|---|---|---|---|
| Actinobacteria | Bacteroidetes | Firmicutes | Proteobacteria |

| Family | | | |
|---|---|---|---|
| Coriobacteriaceae | Bacteroidaceae | Barnesiellaceae | Porphyromonadaceae |
| Prevotellaceae | Ruminococcaceae | Peptostreptococcaceae | Paraprevotellaceae |
| Rikenellaceae | Clostridiaceae | Erysipelotrichaceae | Lachnospiraceae |
| Streptococcaceae | Veillonellaceae | Christensenellaceae | Mogibacteriaceae |
| Enterobacteriaceae | Odoribacteraceae | | |

| Genus | | | |
|---|---|---|---|
| Bacteroides | Parabacteroides | Prevotella | Blautia |
| Faecalibacterium | Butyrivibrio | Streptococcus | Erwinia |
| Clostridium | Coprobacillus | Coprococcus | Dorea |
| Oscillospira | Roseburia | Ruminococcus | Shuttleworthia |
| Moryella | Lachnospira | Adlercreutzia | Holdemania |
| Eubacterium | Christensenella | | |

genera. The overall pattern in this cohort was similar to that in the HMP, although specific values were very different (*Huse et al., 2012*; *Human Microbiome Project Consortium, 2012*). This was probably due to biological variability, including differences in genetics, demographics, and health behavior, as well as technical variability, including differences in sample collection, preparation, sequencing, and bioinformatics processing.

## Demographic and health behavior–related variables shaping the gut microbiota

Because of the wide demographic and health behavior–related variation it captures, the MWRP data set provides a good opportunity to identify associations between these variables and the microbiota. We first performed overall association tests based on α- and β-diversity measures. α- and β-diversity measures provide a holistic view of the microbiota; distinctive yet interrelated, these measures focus on different aspects of microbiota structure. In this study, we chose the following α-diversity measures: the number of observed OTUs (after rarefaction) as a species richness measure, and the Shannon index as an overall diversity measure incorporating both species richness and abundance. In terms of β-diversity, which describes overall microbiota structure, we chose the phylogeny-based unweighted and weighted UniFrac distance metrics; the unweighted UniFrac focuses on community membership, whereas the weighted UniFrac reflects both membership and abundance (*Chen et al., 2012*).

The overall association results are summarized in Tables 3 and 4. Increased BMI was associated with decreased species richness (P = 0.017, Fig. 2A). BMI was also significantly associated with overall microbiota structure, as revealed by both unweighted and weighted UniFrac analysis (P = 0.006 and 0.030, Table 4). Race was associated with both species

**Table 3 Associations (P-values) between demographic and health behavior-related factors and α-diversity measures in the MWRP.**

|  | Age | Sex | BMI | Race | Tobacco | Alcohol |
|---|---|---|---|---|---|---|
| Observed OTUs | 0.463 | 0.062 | **0.017** | **0.013** | 0.381 | **0.040** |
| Shannon index | 0.115 | 0.381 | 0.684 | 0.425 | 0.350 | 0.138 |

**Abbreviations:**
BMI, body mass index; OTU, operational taxonomic unit.

**Table 4 Associations (P-values) between demographic and health behavior-related factors and β-diversity measures in the MWRP.**

|  | Age | Sex | BMI | Race | Tobacco | Alcohol |
|---|---|---|---|---|---|---|
| Unweighted UniFrac | 0.351 | **0.049** | **0.006** | **0.003** | 0.091 | **0.041** |
| Weighted UniFrac | 0.776 | 0.073 | **0.030** | 0.227 | 0.143 | 0.466 |

**Abbreviation:**
BMI, body mass index.

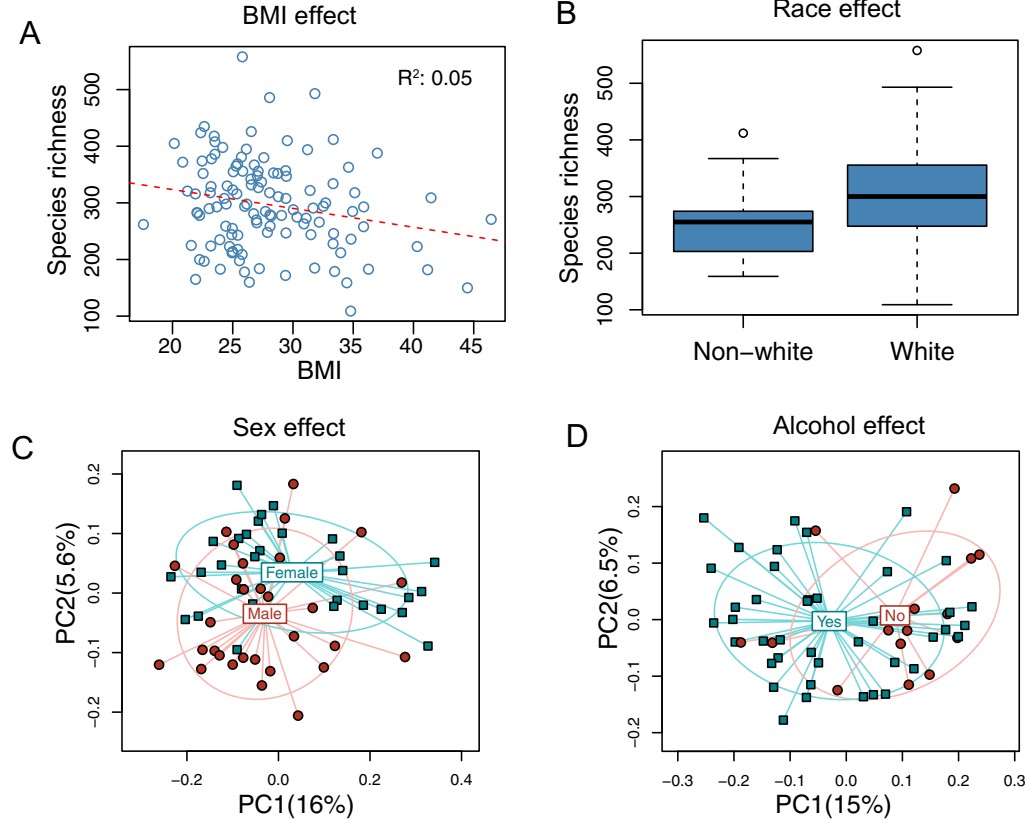

**Figure 2 Associations between demographic and health behavior-related factors and the overall gut microbiota structure.** (A) Increased BMI is associated with decreased species richness (i.e., the observed number of OTUs). (B) Increased species richness was observed in white subjects. The three horizontal lines of the box represent the first, second (median), and third quartiles, respectively, with the whisk extending to the 1.5 interquartile range (IQR). (C) Principal coordinate analysis (PCoA) plot showing a sex effect. (D) PCoA plot showing an alcohol effect. Samples are colored according to group membership, and plotted on axes corresponding the first two principal coordinates (PCs). The percentage of variability explained by each PC is indicated in the parentheses.

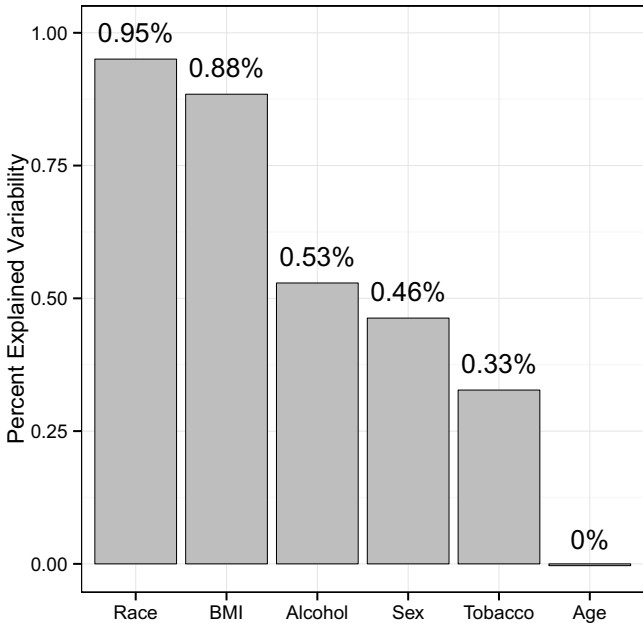

**Figure 3 Percentage of variability explained by the demographic and health behavior-related factors.** The unweighted UniFrac distance was used to summarize the microbiota variability. The distance-based $R^2$ was adjusted to reduce inflation due to a small sample size.

richness (P = 0.013, Fig. 2B) and overall microbiota structure (P = 0.003, unweighted UniFrac), with white subjects exhibiting greater species richness. Sex and alcohol use were both associated with overall structure (P = 0.049 and 0.041 respectively, unweighted UniFrac, Figs. 2C and 2D). In addition, alcohol use was associated with increased species richness (P = 0.040, Table 3). Some evidence of association of tobacco use with overall microbiota structure was observed (P = 0.091, unweighted UniFrac). Age, by contrast, was not significantly associated with measures of α- or β-diversity. Statistical significance was primarily observed for unweighted measures, such as species richness and the unweighted UniFrac distance, indicating that these demographic and health behavior–related factors chiefly affect community membership or the rare biosphere of the microbiota (*Chen et al., 2012*).

Next, we quantified the effect sizes of these variables with adjusted, distance-based $R^2$ values (see Methods). Using the unweighted UniFrac distance to summarize overall microbiota variability, the adjusted $R^2$ values were 0.95% for race, 0.88% for BMI, 0.53% for alcohol use, 0.46% for sex, 0.33% for tobacco use, and 0.0% for age (Fig. 3), indicating huge intersubject variability relative to the variability associated with these factors.

## Microbial signature of demographic and health behavior–related factors

We next set out to identify the microbial signature of the demographic and behavioral factors that were most significantly associated with overall microbiota structure (BMI, sex, race, tobacco use, and alcohol use). We used permutation-based univariate association

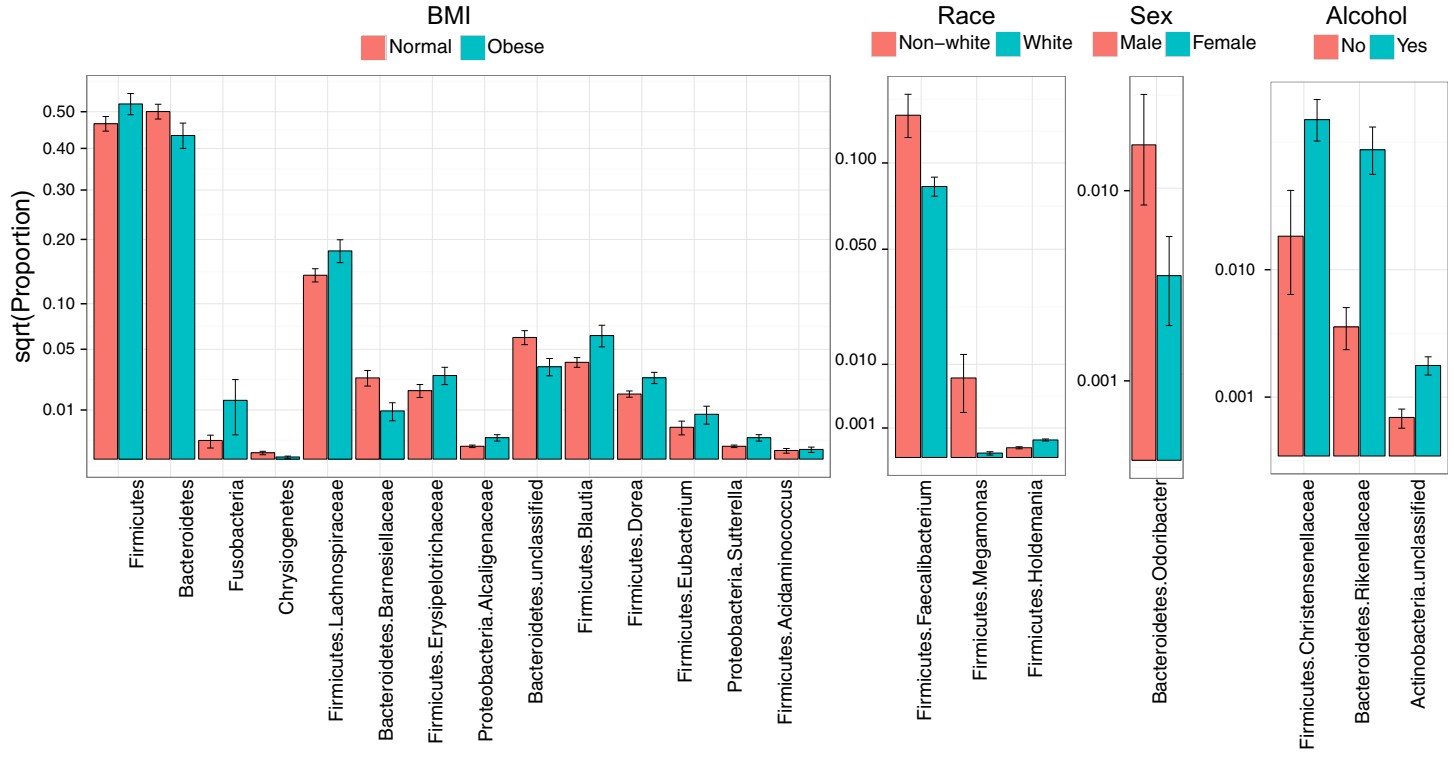

**Figure 4 Microbial signatures of demographic and health behavior-related factors: BMI, race, sex, and alcohol use.** Barplots show the mean relative abundance and standard error of bacterial taxa in each subject subgroup. Taxa were selected based on univariate association tests with an FDR of 10%. Here, we have discretized BMI into normal and obese groups based on the cutoff of 30 kg/m$^2$ for visualization purposes. However, we treated BMI as a continuous outcome in our association tests.

tests to identify associated bacterial taxa at the phylum, family, and genus level. FDR control was used to correct for multiple testing.

Consistent with overall association tests, a number of bacterial taxa were associated with BMI at an FDR of 10% (Fig. 4, Table 5). In particular, we observed an increase in Firmicutes and a decrease in Bacteroidetes in obese subjects, consistent with findings from some previous studies (*Turnbaugh et al., 2009*; *Walters, Xu & Knight, 2014*). In addition, we detected an increase in the genus *Eubacterium*, consistent with two previous studies that reported associations between *Eubacterium rectale* and obesity (*Walters, Xu & Knight, 2014*). Interestingly, we also found an increase in Fusobacteria, which have been shown to be associated with colon cancer (*Rubinstein et al., 2014*).

Race, sex, and alcohol use were also associated with changes in the abundance of specific taxa (Fig. 4, Table 5); here, we focus only on the most statistically significant associations (Q-value < 0.05). Relative to non-whites, whites had higher levels of *Holdemania* and lower levels of *Megamonas* (Table 5). Interestingly, although *Holdemania* is one of the 30 most abundant genera in the gut microbiome of individuals from high-income countries (*Arumugam et al., 2011*), it was absent in a recent study of children from Bangladesh. It is not clear why race might be associated with the abundance of this common but not universal bacterial genus. *Megamonas* species are associated with colon cancer (*Weir et al., 2013*), but they are also associated with normal glucose tolerance

**Table 5  Bacterial taxa associated with BMI, race, sex, and alcohol use based on univariate associations at an FDR of 10%.**

| BMI | Mean proportion | | Log2 Fold Change | Q value |
|---|---|---|---|---|
| | Obese | Normal | | |
| | Phylum | | | |
| Bacteroidetes | 4.339E-01 | 5.004E-01 | −0.206 | 0.080 |
| Firmicutes | 5.226E-01 | 4.660E-01 | 0.165 | 0.080 |
| Chrysiogenetes | 1.927E-05 | 1.680E-04 | −3.124 | 0.080 |
| Fusobacteria | 1.433E-02 | 1.446E-03 | 3.309 | 0.080 |
| | Family | | | |
| Bacteroidetes; Barnesiellaceae | 9.637E-03 | 2.734E-02 | −1.505 | 0.098 |
| Firmicutes; Erysipelotrichaceae | 2.898E-02 | 1.942E-02 | 0.578 | 0.078 |
| Firmicutes; Lachnospiraceae | 1.796E-01 | 1.401E-01 | 0.358 | 0.098 |
| Proteobacteria; Alcaligenaceae | 1.909E-03 | 6.867E-04 | 1.475 | 0.098 |
| | Genus | | | |
| Bacteroidetes; unclassified | 3.534E-02 | 6.134E-02 | −0.795 | 0.049 |
| Firmicutes; Acidaminococcus | 3.915E-04 | 3.081E-04 | 0.346 | 0.098 |
| Firmicutes; Blautia | 6.326E-02 | 3.883E-02 | 0.704 | 0.049 |
| Firmicutes; Dorea | 2.743E-02 | 1.757E-02 | 0.643 | 0.049 |
| Firmicutes; Eubacterium | 8.339E-03 | 4.210E-03 | 0.986 | 0.049 |
| Proteobacteria; Sutterella | 1.909E-03 | 6.867E-04 | 1.475 | 0.098 |
| Race | Mean proportion | | Log2 Fold Change | Q value |
| | Non-white | White | | |
| | Genus | | | |
| Firmicutes; Faecalibacterium | 1.351E-01 | 8.466E-02 | 0.674 | 0.098 |
| Firmicutes; Holdemania | 1.092E-04 | 3.542E-04 | −1.698 | 0.049 |
| Firmicutes; Megamonas | 7.288E-03 | 2.226E-05 | 8.355 | 0.049 |
| Sex | Mean proportion | | Log2 Fold Change | Q value |
| | Yes | No | | |
| | Genus | | | |
| Bacteroidetes; Odoribacter | 1.344E-02 | 4.606E-03 | 1.545 | 0.067 |
| Alcohol intake | Mean proportion | | Log2 Fold Change | Q value |
| | Yes | No | | |
| | Family | | | |
| Bacteroidetes; Rikenellaceae | 2.702E-02 | 4.803E-03 | 2.492 | 0.002 |
| Firmicutes; Christensenellaceae | 3.260E-02 | 1.393E-02 | 1.227 | 0.030 |
| | Genus | | | |
| Actinobacteria; unclassified | 2.362E-03 | 4.288E-04 | 2.462 | 0.062 |

(vs type 2 diabetes) (*Zhang et al., 2013*). Again, the connection between *Megamonas* and race is unclear. Finally, current alcohol users had lower levels of the family Rikenellaceae, which is also depleted in end-stage liver disease (*Bajaj et al., 2014*), and higher levels of Christensenellaceae (Table 5), a bacterial family enriched in individuals with low BMI

(*Goodrich et al., 2014*). As is true for race, the mechanisms linking sex and alcohol use with changes in these taxa are not yet clear. We were unable to identify specific taxa associated with tobacco use after FDR control.

## DISCUSSION

In this study, we present the MWRP, a reference cohort available for researchers seeking to identify gut microbiome biomarkers related to health or disease. This cohort encompasses a wider range in age and BMI than traditional "healthy" cohorts and contains data on important demographic and health behavior–related variables, such as sex, race, tobacco use, and alcohol use. Consistent with previous studies, we observed large intersubject variability in taxonomic abundance and a small number of core taxa in this cohort (*Huse et al., 2012*; *Human Microbiome Project Consortium, 2012*). Moreover, our finding that most of the demographic and health behavior–related variables we investigated were associated with changes in the gut microbiota underscores the importance of having such a diverse cohort available for study. The MWRP can be used to reduce confounding and increase reproducibility in future studies of the gut microbiome that take place in the Midwestern United States.

Within the "normal" cohort of the MWRP, potentially important associations between demographic, health, and behavioral variables and the microbiome emerged. We found an increase in Firmicutes and a decrease in Bacteroidetes in obese subjects; moreover, BMI was associated with decreased species richness as well as overall microbiota structure, confirming the results of previous studies on this subject (*Ley et al., 2006*; *Turnbaugh et al., 2006*; *2009*; *Verdam et al., 2013*; *Sepp et al., 2014*). Health-related behaviors, such as alcohol and tobacco use, are difficult to measure, so they are infrequently studied. Although less attention has been paid to these factors, they nonetheless have been linked to profound changes in the fecal microbiota at the genus level (*Biedermann et al., 2013*; *Mutlu et al., 2014*). In our study, alcohol use was associated with overall microbiota structure, and a trend between alcohol use and increased species richness was also detected. Interestingly, alcohol use was associated with an increase in bacteria of the Rikenellaceae family, which are depleted in end-stage liver disease (*Bajaj et al., 2014*), a potential consequence of alcoholism. Some evidence of association between tobacco use and overall microbiome structure was also observed; given a larger sample size or more detailed data on smoking habits, a significant association between the two may emerge. Many previous studies have found a significant effect of age on the gut microbiota (*Hopkins, Sharp & Macfarlane, 2001*; *Hébuterne, 2003*; *Mäkivuokko et al., 2010*). However, we did not find an association between age and $\alpha$- or $\beta$-diversity measures in this study.

The differences in associations between demographic and health behavior–related variables and weighted and unweighted UniFrac distances are potentially meaningful. The unweighted UniFrac distance measures the presence or absence of particular bacterial taxa and is a more qualitative measure of $\beta$-diversity than the weighted UniFrac. The unweighted UniFrac is more sensitive to rare species within a microbial community, as a species will appear absent when its abundance falls below the detection limit of the
sequencing machine (*Chen et al., 2012*). As such, a more significant association with unweighted UniFrac distance (as was found for BMI, sex, alcohol use, race, and tobacco use) may indicate that changes in rare bacteria are altering the community structure. This assertion is supported by our α-diversity analysis; demographic and health behavior–related variables were more significantly associated with community richness, which is based on presence/absence data for OTUs, than overall diversity based on abundance data (i.e., the Shannon index). Finally, in this study, the majority of taxa associated with these variables were rare. Taken together, the results of these separate, yet interrelated, diversity analyses indicate that demographic and health behavior–related factors may affect primarily rare bacterial species within the gut. However, BMI and sex show significant or marginally significant associations with the weighted UniFrac, which is more sensitive to abundant lineages, indicating that some abundant bacterial lineages may also be affected by these factors. Regardless of the analytic tool used, our data suggest that microbial communities are altered by demographic and health behavior–related factors.

The implications of the extensive associations identified in this study are threefold. First, demographic and health behavior–related factors may confound comparative analyses if their distribution differs between groups of interest. These variables must either be matched across groups or adjusted for in statistical models. Importantly, statistical adjustment becomes insufficient when the range of variables differs between groups, as statistical models are less good at extrapolation than interpolation. Nonlinear effects can also impede statistical adjustment. In these cases, sample matching is more appropriate for reducing potential confounding. The MWRP thus provides a good reference panel with which to perform sample matching based on demographic and health behavior–related factors, as it encompasses a wide range for each variable. Second, even if demographic factors do not confound an analysis (i.e., the distribution of these factors is similar between groups), adjustment for these independent predictors in a model will improve statistical power by reducing random error. In some scenarios, significant associations may only be revealed when demographic factors are accounted for. Lastly, other factors important in the Midwest that are difficult to measure in a study like this one, such as dietary habits, may exist. If not well controlled, a case-control–based microbiome biomarker study may capture associations due to these unappreciated confounders, reducing reproducibility. By providing a representative reference sample drawn from the same geographic region that cases are drawn from, the MWRP will be instrumental in minimizing interference by these potential hidden confounders in studies of the microbiome that take place in the Midwest.

Although it is reassuring that emerging and consistent trends exist in findings reported by diverse labs for various conditions, such as BMI, our research has identified frailties that need further attention if we are to achieve clinical-grade results in microbiome studies. Despite standardized protocols, the implementation of robotic equipment, and uniform technical support, we found evidence of batch effects in this study. We adjusted batch effects in the statistical model to improve statistical power as well as to reduce potential confounding effects (*Leek et al., 2010*). When batch effects confound a variable

of interest, much power will be lost due to batch adjustment. In this case, we suspect that batch effects, which were entangled with age, may have prevented us from detecting some of the associations between age and the microbiome that have been reported in other studies. The clinical utility of microbiome sequencing data may be constrained unless this effect can be overcome. To minimize the impact of batch effects, we advise sample randomization, rigorous and standardized collection methods, and the coordination of sample processing and sequencing for research and biomarker development. For clinical sample management, we propose standardized collection and processing protocols, robotics where applicable, and internal controls.

The MWRP also allows us to estimate effect sizes for demographic and health behavior–related factors, which is fundamental to moving forward with human microbiome studies that use the stool to study the gut microbiome. It is important to consider factors with larger effect sizes in the study design phase, whereas factors with smaller effect sizes are of less concern. Effect sizes can help prioritize factors when matching subjects from a disease group with samples from a control group. Although $\beta$-diversity association P-values may reflect effect sizes, the coefficient of determination, $R^2$ (i.e., the percent of variation in an outcome variable explained by a factor of interest), is a more suitable measure (*McArdle & Anderson, 2001*). Using unweighted UniFrac distances to summarize overall microbiota variability, adjusted $R^2$ values were 0.95% for race, 0.88% for BMI, 0.53% for alcohol use, 0.46% for sex, 0.33% for tobacco use, and 0.0% for age. The small values of $R^2$ may be partially due to noise from sample collection, preparation, and sequencing, but they also indicate that these factors explain only a small fraction of the large intersubject variability of the gut microbiota. Although small in effect size, these factors may still be important confounders, as many of the associations we find in public health also have very small effect sizes. Race and BMI, which have the largest $R^2$ values, warrant special attention in study design.

The diverse associations between demographic and health behavior–related factors and the gut microbiota, and their small effect sizes, have motivated us to expand the list of demographic and behavioral variables investigated in the future, with the goal of identifying more factors that shape the gut microbiota; indeed, our next step is to expand the MWRP sample to explore factors such as diet, exercise, depression, and anxiety. A larger sample, coupled with more detailed documentation of variables, will be instrumental for addressing potential confounding due to factors that are rarely represented in the normal population but are enriched in groups with diseases, as well as for identifying the microbiota features that best distinguish between diseased and healthy states. In addition, we will extend our current taxonomic profiling to functional profiling, to characterize a possible functional core in the MWRP. Longitudinal studies of the microbiome, which are conducive to establishing a mechanistic link between the microbiota and a phenotype, will also be pursued. Finally, similar studies of the impact of demographics on microbiomes from other body sites will be conducted to facilitate microbiome biomarker discovery for a broader range of diseases.

## CONCLUSIONS

In conclusion, the MWRP will be instrumental in elucidating demographic and health behavior–related factors impacting the human gut microbiota, as well as increasing reproducibility in microbiome biomarker discovery. It represents a key step in translating microbiome discoveries into clinical applications and, ultimately, in improving patient care.

### Funding

Funding was received from the Center for Individualized Medicine at the Mayo Clinic (http://mayoresearch.mayo.edu/center-for-individualized-medicine/) and the National Institutes of Health (NIH) (1R01CA179243). The funders had no role in study design, data collection and analysis, decision to publish, or preparation of the manuscript.

### Grant Disclosures

The following grant information was disclosed by the authors:
National Institute of Health: 1R01CA179243.

### Competing Interests

Jun Chen is an Academic Editor for PeerJ.

### Author Contributions

- Jun Chen analyzed the data, wrote the paper, prepared figures and/or tables, reviewed drafts of the paper.
- Euijung Ryu contributed reagents/materials/analysis tools, reviewed drafts of the paper.
- Matthew Hathcock contributed reagents/materials/analysis tools, reviewed drafts of the paper.
- Karla Ballman contributed reagents/materials/analysis tools, reviewed drafts of the paper.
- Nicholas Chia conceived and designed the experiments, performed the experiments, analyzed the data, reviewed drafts of the paper.
- Janet E Olson contributed reagents/materials/analysis tools, reviewed drafts of the paper.
- Heidi Nelson conceived and designed the experiments, wrote the paper.

### Human Ethics

The following information was supplied relating to ethical approvals (i.e., approving body and any reference numbers):
Mayo Clinic IRB #13-003694

# PeerJ

## Data Deposition
BioProject accession number: PRJNA297510

## Supplemental Information
Supplemental information for this article can be found online at http://dx.doi.org/10.7717/peerj.1514#supplemental-information.

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
