# Peer review of "Impact of demographics on human gut microbial diversity in a US Midwest population"

_PeerJ, doi:10.7717/peerj.1514_

## Round 0.1 · original submission · Minor Revisions

Both reviewers appreciate the significance of the study and study findings. There are some concerns that they want the authors to address or clarify. Majority of them are minor presentation issues. Please pay attention to the first suggestion that Reviewer 1 brought up.

Reviewer 1 ·

Basic reporting

Please see General Comments.

Experimental design

Please see General Comments.

Validity of the findings

Please see General Comments.

Additional comments

Comments on the PeerJ manuscript “Impact of demographics on human gut microbial diversity in a US Midwest population”

In this manuscript, the authors analyze the relationships between diversity (alpha and beta) of gut microbiota and different health behavior-related factors including sex, age, race, body mass index, alcohol use, and the tobacco use. I think the topic is interesting and especially of importance to evaluate the differences/changes of gut microbiota for different disease cases. Due to my backgrounds, I can only offer thoughts and comments from a statistical perspective. I hope my comments are useful for the authors in preparing a revision.

(1) My major comment is about the diversity measures used in the analyses. For example, the authors chose two alpha diversity measures: the number of observed OTUs (after rarefaction) as a species richness measure, and the Shannon index as an overall diversity measure incorporating both species richness and abundance. It is understandable that the authors use a rarefied observed OTUs because the observed OTUs strongly depends on the sample size and sample completeness. However, the observed Shannon index also depends on sample size and sample completeness. I would suggest the authors read the following paper for rarefaction and extrapolation of species richness and Shannon diversity measure:
Chao, A., Gotelli, N. G., Hsieh, T. C., Sander, E. L., Ma, K. H., Colwell, R. K. and Ellison, A. M. (2014). Rarefaction and extrapolation with Hill numbers: a framework for sampling and estimation in species biodiversity studies. Ecological Monographs 84, 45-67.
In the above paper, Chao et al. also pointed out that a sample of a given size may be sufficient to fully characterize a low-diversity assemblage, but insufficient to characterize a rich-assemblage. Thus, when the species counts of two equally-large samples are compared, one might be comparing a nearly complete sample to a very incomplete one. This will generally lead to underestimate the difference in diversity between the sites. Chao et al. also proposed standardization by sample completeness rather than sample size. I would suggest the authors consider the approach to standardizing sample completeness.

(2) In the analyses, the authors exclude all singletons and use only the observed OUTs in the non-singletons. I can understand this is due to possible sequencing errors for low-frequency counts, producing spurious singletons. In a recent preprint (Chiu and Chao 2015 https://peerj.com/preprints/1353.pdf), they proposed an estimator of the true singleton count in terms of the frequency counts of doubletons, tripletons and quadrupletons. In other words, statistical method can help estimate the true singleton count. Based on estimated singleton count, researchers can make more meaningful rarefaction and extrapolation method (see the preceding paragraph) and can infer a lower bound about the number of undetected OTUs, which may provide useful information in analysis.

(3) The authors found that increased BMI has a significance effect on species richness (P= 0.025). However, the coefficient of determination R^2 is only 0.05, meaning that BMI can only explain 5% of the variation of species richness. With such a low value of R^2, I would think the model or the relationship is not an adequate model. The low value of R^2 implies that more covariates (regressors) or explanatory variables should be added to the model. From a perspective of statistical theory, I would suspect that this statistical significance is most likely due to a large sample size, rather than due to biological significance. Similar comments apply to other significant results.

Reviewer 2 ·

Basic reporting

Please see below

Experimental design

Please see below

Validity of the findings

Please see below

Additional comments

This paper considers a US Midwest population and examines the impact of demographic variables on the human gut microbial diversity. The topic is interesting and scientifically important. The paper is well-written. I have the following comments and suggestions:
1. In lines 105—106, it is apparent that a younger cohort was selected in 2013 and an older one in 2014. Is there a specific reason for that? In this study with only 118 samples, how is the distribution of the “age” variable? Do you use it as a binary variable with 50-year-old as the cut-off?
2. After the “sample preparation and sequencing” and before the “statistical analyses”, could you add the part on how to create OTU tables? And discuss how sensitive the method is?
3. The batch effect is considered in the statistical analysis, however, the batch effect is not that well discussed or addressed in the final conclusion/discuss part. How important do you think the batch effect in your study, and how to generally handle the batch effect in microbiome studies?
4. In line 224, could you cite what are the “previous studies”?
5. In the statistical analysis assessing the association with OTU counts, how do you address the low-abundance OTU? I’m a little bit confusing on the “outcome was approximately normal” in lines 168—169, and “the non-normality of the taxa data”. The relative abundance analysis was part of the alpha-diversity association, or not?
6. You mentioned “tobacco use shows a trend of association with the microbiota”. Did you perform a trend test for this?
7. With respect to the association with demographics, do you think is there any implications to other microbiome studies, like oral microbiome?
8. In line 468, you missed the name of the journal’s title.

---

## Round 0.2 · accepted · Accept

Happy Thanksgiving! Both reviewers think you have addressed their previous concerns and the current version is ready for publication. It will be a nice addition to the literature.

Reviewer 1 ·

Basic reporting

No comments

Experimental design

No comments

Validity of the findings

No comments

Additional comments

In this revision, I think the authors have properly addressed my comments/suggestions. I am satisfied with the revision and recommend acceptance.

Reviewer 2 ·

Basic reporting

NONE

Experimental design

NONE

Validity of the findings

NONE

Additional comments

NONE